# Physics-Informed, Data-Driven Model for Atmospheric Corrosion of Carbon Steel Using Bayesian Network

**DOI:** 10.3390/ma16155326

**Published:** 2023-07-28

**Authors:** Taesu Choi, Dooyoul Lee

**Affiliations:** Department of Weapon System, Korea National Defense University, Nonsan 33021, Republic of Korea; winnercts92@gmail.com

**Keywords:** model calibration, Bayesian network, atmospheric corrosion, carbon steel

## Abstract

Atmospheric corrosion is a significant challenge faced by the aviation industry as it considerably affects the structural integrity of an aircraft operated for long periods. Therefore, an appropriate corrosion deterioration model is required to predict corrosion problems. However, practical application of the deterioration model is challenging owing to the limited data available for the parameter estimation. Thus, a high uncertainty in prediction is unavoidable. To address these challenges, a method of integrating a physics-based model and the monitoring data on a Bayesian network (BN) is presented herein. Atmospheric corrosion is modeled using the simulation method, and a BN is constructed using GeNie. Moreover, model calibration is performed using the monitoring data collected from aircraft parking areas. The calibration approach is an improvement over existing models as it incorporates actual environmental data, making it more accurate and applicable to real-world scenarios. In conclusion, our research emphasizes the importance of precise corrosion models for predicting and managing atmospheric corrosion on carbon steel. The study results open new avenues for future research, such as the incorporation of additional data sources to further improve the accuracy of corrosion models.

## 1. Introduction

Atmospheric corrosion is a time-dependent deterioration mechanism that significantly affects the structural integrity of equipment operated for long periods. According to Findlay and Harrison [1], more than 20% of aircraft component failures are related to corrosion. Corrosion accelerates the aging of aircraft structures by interacting with other deterioration mechanisms, such as fatigue cracking and stress corrosion cracking [2,3]. Factors contributing to the corrosion of carbon steel, which is one of the aircraft structural materials, are the type of metal, electric potential difference, atmospheric corrosion environment, etc. [4,5,6]. Because aircraft operate in external environments for extended periods, they are greatly affected by atmospheric corrosion [4,7]. ISO 9223 identifies relative humidity, chloride deposition rate, and sulfur dioxide deposition rate as the main factors that affect material corrosion in atmospheric environments [8].

To prevent corrosion, an appropriate corrosion deterioration model is necessary. However, because of the limited availability of data for the parameter estimation and the high level of uncertainty in predicting results, a practical application of the corrosion deterioration model is difficult [9,10,11]. Previous studies have modeled multiple physical phenomena related to atmospheric corrosion using commercial software such as COMSOL Multiphysics (4.2 and 5.6) [12,13,14,15,16,17]. The previous models included several assumptions and had limitations in representing the corrosion environment of specific regions, because the modeling results were verified by conditions created in a laboratory.

This study proposes a method of combining a physics-based model with a data-driven model. The combined model can provide a metamodel for risk analysis. Section 2 describes the preparation process for obtaining an improved atmospheric corrosion environment model. Moreover, that section presents a procedure for the real-world experimental measurement of atmospheric corrosion and a simulation scheme for the physics-based model. A calibration method using a Bayesian network (BN) and the overall research procedure are also described. Section 3 presents and discusses the results of atmospheric corrosion monitoring. That section also discusses the simulation results and the changes in the calibrated model using BN. Finally, Section 4 presents the conclusions and limitations of the study, as well as suggestions for future research directions.

## 2. Materials and Methods

### 2.1. Definition of Outdoor Environment

The corrosion characteristics of aircraft structures can vary depending on the outdoor environment they are exposed to. Therefore, in this study, the outdoor environment was classified using the following three criteria: (1) coastal vs. inland, (2) revetment vs. shelter, and (3) roof installed vs. uninstalled. Data measurements were conducted based on these criteria, and atmospheric corrosion environment modeling and calibration were performed.

### 2.2. Atmospheric Corrosion Monitoring

In reference to a United States Air Force case study [18], a set of atmospheric-corrosion-monitoring equipment and specimen cards were designed to measure the corrosion rate, relative humidity, and chloride deposition rate of key metallic materials (Figure 1). The specimen cards comprised six metallic specimens, including silver (Ag) to measure the chloride deposition rate; copper (Cu); aluminum alloys (AA2024, AA6061, and AA7075); and carbon steel for corrosion rate monitoring. The specimen cards were installed in aircraft parking areas at 13 air force bases nationwide and were retrieved and analyzed by the Aero Technology Research Institute [7].

The retrieved specimens were cleaned to remove foreign substances, and the corrosion rate was calculated according to ASTM G1 [7,19]. For carbon steel specimens, the weight before and after cleaning was measured. The specimen was repeatedly rinsed with a 50 vol% hydrochloric acid solution for 2 min, and the weight was measured after each rinse. The mass loss due to corrosion was estimated by determining the point at which the slope of the graph changed because of the difference in the degree of reaction between the corrosion products and the substrate.

The relative humidity was measured using a temperature and humidity sensor (Testo-174H). Further, the time of wetness (TOW), which is the duration when the temperature exceeds 0 °C and the relative humidity exceeds 80%, was calculated. The annual chloride deposition rate was estimated by an X-ray photoelectron spectroscopy (XPS) analysis of the sample surface [7]. To determine the mass of each element accumulated on the silver specimen exposed for one year, the etching rate was calculated as follows.
(1)$$Etch\;Rate=\left(\frac{Yw}{\rho }\right)\left(\frac{I}{A}\right)\times 10^{-2}$$
where *Y* represents the sputtering yield, *w* is the atomic mass, *ρ* is the density, *I* is the ion beam current (in μA), and *A* represents the etching area (in mm^2^).

Specifically, the etching depth was calculated by multiplying the exposure time of the ion beam by the etching rate. The etching volume was calculated by multiplying the etching depth by the etching area, and the etching mass was calculated by multiplying the silver density by the etching volume. Finally, the chloride deposition rate during the exposure period was calculated using the vertical composition and distribution analysis (depth profiling) results of each element. The calculation method involved some errors due to the variation in composition with changing depth in the sample. However, it was deemed valid as it showed a relatively high correlation with the chloride deposition rate of the same sample measured using other quantitative methods, such as X-ray diffraction and coulometric reduction [7,20].

### 2.3. Atmospheric Corrosion Modeling

Tsoutsani [21] presented an atmospheric corrosion model in one dimension using a simulation method. Accordingly, we followed the simulation method, as shown in Figure 2. Boundary 1 represents the flux boundary condition due to electrochemical reactions occurring on the iron surface (electrode surface), and boundary 2 was set as the constant electrolyte potential. We assumed the exclusion of the influence of CO_2_ in water in this physical model. The following reactions were considered to occur on the electrode surface:(2)$$Fe\to Fe^{2+}+2e^{-}$$
(3)$$2H^{+}+2e^{-}\to H_{2},$$
(4)$$O_{2}+2H_{2}O+4e^{-}\to 4OH^{-}.$$

Equation (2) represents the oxidation of iron, and Equations (3) and (4) are the hydrogen evolution and oxygen reduction reactions, respectively.

To implement electrochemical reactions in the model, the secondary current distribution interface was utilized. This interface explains the activation loss caused by reactions at the electrode surface. The relationship between charge transfer and overpotential can be expressed using Equations (5) and (6).

The corrosion current density can be expressed by the Butler–Volmer equation:(5)$$i_{corr}=i_{0}\left(\exp\left(\frac{\alpha_{\alpha }F\eta }{RT}\right)-\exp\left(\frac{{-\alpha }_{c}F\eta }{RT}\right)\right)$$
where *i_corr_* represents the corrosion current density; *i*_0_ is the initial current density; *α_α_* and *α_c_* are the forward (anodic) and backward (cathodic) transfer coefficients, respectively; *F* is the Faraday constant (96,485 C/mol); *R* is the gas constant (8.314 J/Kmol); and *T* is the temperature. Here, *η* represents the activation overpotential and can be expressed using the Tafel equation:(6)$$\eta =\beta \;log\left(\frac{i}{i_{corr}}\right)$$
where *β* is the Tafel slope (unit: voltage), and *i* is the current density.

In summary, the secondary current distribution interface is used to calculate the current and potential distribution in an electrochemical cell, assuming that the electrolyte layer has a constant electrical conductivity. Therefore, the charge transfer satisfies Ohm’s law. Under the assumption that the conductivity is constant, the changes in the electrolyte composition due to electrochemical reactions can be ignored, and the ion movement can be considered to contribute only to changes in the current in the electrolyte [21].
(7)$$Current=\frac{Electric\;field}{Resistance}$$

Mizuno, Shi, and Kelly [13] established an atmospheric corrosion model considering the changes in the electrical conductivity and thickness of the electrolyte layer due to the relative humidity and salt load density. Therefore, we considered these theoretical assumptions for the simulation. In particular, the simulation allowed us to obtain a physics-based model for atmospheric corrosion.

### 2.4. Calibration Using Bayesian Network

BNs are probabilistic models represented by directed acyclic graphs, which model the joint probability mass function (PMF), p(x), of a set of random variables X. As the number of variables in X increases, the space of X, i.e., the number of outcome states for which p(x) must be computed, increases exponentially. However, BN modeling enables an efficient computation by representing the joint probability distribution as a product of local (conditional) distributions for each variable.

An expectation–maximization (EM) algorithm is commonly utilized for inferring parameters when the available data are incomplete. The EM algorithm comprises two steps: an E-step and an M-step. In the E-step, an estimate of the probability distribution over the possible missing data is computed using the current or previously estimated parameters. This is achieved by selecting a function, *g_t_*, which reduces the objective function log Pr(*x; z*) at all points; *z* is a vector of the unknown parameters. In the M-step, the maximum likelihood method is employed to determine the local maximum of *g_t_*. The two steps are repeated until the parameters converge, yielding the global maximum of the objective function. Because the objective function is equivalent to *g_t_*, the following relationship holds true [22]:(8)$$\log\mathrm{Pr}\left(x;{\hat{z}}^{\left(t\right)}\right)=g_{t}\left({\hat{z}}^{\left(t\right)}\right)\leq g_{t}\left({\hat{z}}^{\left(t+1\right)}\right)=\log\mathrm{Pr}\left(x;{\hat{z}}^{\left(t+1\right)}\right).$$

Accordingly, the objective function consistently increases with each iteration of the EM algorithm. In this study, the parameters of the physics-based model (regression model) were updated using the EM algorithm.

### 2.5. Framework for the Study

The research methodology comprised two main steps, as shown in the flowchart in Figure 3. First, a physics-based model for atmospheric corrosion environment was developed. The implemented model was then parameterized into a nonlinear model for the TOW and chloride deposition rate. In this nonlinear model, we assumed that the chloride deposition rate followed a power-law function, and the TOW followed an exponential function, referring to the relationship equation for corrosion rate in ISO 9223 [8]. Here, the model was represented by parameters (*θ*_0_, *θ*_1_, and *θ*_2_). Subsequently, we performed a logarithmic transformation (linear model) on this nonlinear model for the convenience of calculations. Model calibration was performed using the transformed model and the monitoring data obtained from the study by Lee et al. [7]. The calibration procedure involved constructing dose–response functions using the BN. *θ_0_*, *θ_1_*, and *θ_2_* were updated during the calibration process.

## 3. Results and Discussion

### 3.1. Atmospheric Corrosion Monitoring

Kwon and Lee [23] distinguished the monitoring data to establish an algorithm for determining aircraft wash intervals. They considered that the geographic environment was divided into coastal and inland areas and also took into account whether it was sheltered or not. In Table 1, bases with a subscript *R* (${UC}_{1R}$, ${UC}_{2R}$, ${UI}_{2R}$, and ${UI}_{4R}$) have roofs installed over their revetments. The revetment of base ${UC}_{1R}$ has a high roof installed.

Choi, Lee, and Bahn [20] performed a quantitative analysis of the chloride deposition rate on silver in the atmospheric corrosion environment in Korea. They calculated the chloride concentration (in at%) as a function of etching time for each region through an XPS depth-profile analysis.

The monitoring data for the atmospheric corrosion environment, including distance from the coast, TOW, chloride deposition rate, and carbon steel corrosion rate, are presented in Figure 4. The TOW and carbon steel corrosion rate exhibit a positive correlation (Pearson correlation coefficient of 0.6), while the chloride deposition rate and carbon steel corrosion rate show a strong correlation (Pearson correlation coefficient of 0.9).

### 3.2. Atmospheric Corrosion Modeling

The results of the simulations are shown in Figure 5. The physics-based model was obtained by the linear regression of simulation results. To obtain a linear regression model, we used the Latin hypercube technique to generate samples for the TOW and the chloride deposition rate. In this regard, considering the relationship between the measured chloride deposition rate [7] and chloride concentration [20], we assumed a range of chloride ion concentration from 0 to 42 at% and a limit concentration of 100 at%. Subsequently, we determined the corrosion rate values through calculations using the physics-based model and training samples obtained from the simulations. Accordingly, we could acquire the metadata set through this procedure. Using the metadata and logarithmic transformation of the chloride deposition rate, we obtained a linear regression model. From the linear regression model, the annual corrosion rate of the physics-based model was found to be proportional to the TOW and the chloride deposition rate.

### 3.3. Calibration and Validation

Figure 6 illustrates the calibration process to implement a dose–response function and estimate the parameters accordingly. The prior distributions of the parameters (*θ_0_*, *θ_1_*, and *θ_2_*) can be obtained from the physics-based model. Performing a linear regression on the physics-based model with a logarithmic transformation yielded the mean and standard errors for the parameters. Discretizing these variables enabled the construction of prior distributions for the BN appropriately. The parameters were assumed to follow a normal distribution. During the calibration process, the posterior distributions for the parameters were obtained using the data from Figure 5 and the EM algorithm.

The posterior distributions were significantly different from the prior distributions (Table 2). Figure 7a shows a significant difference between the simulation results (“O” marker) and the monitoring data (“X” marker). In this case, the simulation results were used to generate a prior distribution, while the monitoring data were used to update this prior distribution and generate a posterior distribution. This means that the posterior distributions of each parameter shifted due to the inclusion of the monitoring data.

log(*θ*_0_) represents the intercept of the model’s logarithmically transformed corrosion rate axis. Overall, the monitoring data were higher than the simulation data; therefore, log(*θ*_0_) increased. *θ*_1_ represents the rate of change of the corrosion rate (log scale) with respect to the chloride deposition rate (log scale). In the monitoring data, the rate of change with respect to the chloride deposition rate was lower than that in the simulation data. Therefore, the posterior distribution of *θ*_1_ shifted to smaller values compared to the prior distribution. Finally, *θ*_2_ represents the rate of change of the corrosion rate (log scale) with respect to the TOW, and it can be observed that its value increased based on the monitoring data (Figure 7a).

The goodness of fit of the improved model, incorporating the updated parameters, improved significantly, as indicated by the reduction in the root-mean-squared errors (RMSEs) from 4.12 (in the prior distribution) to 0.688 (in the posterior distribution). When the corrected model was fitted back to the original function through an inverse logarithmic transformation, it resulted in Figure 7b.

### 3.4. Discussion

The results of the comparison between the fitness of the models and the monitoring data are shown in Table 3. The logarithmically transformed models were compared with the monitoring data that had also undergone a logarithmic transformation. It can be observed that the calibrated log-transformed model significantly improved the sum-of-squares error and RMSE. As a result, the RMSE of the physical model improved from 0.392 to 0.308.

According to ISO 9223 [8], the corrosion rate calculation formula for carbon steel is given as follows:(9)$$r_{corr}=0.3592\;P_{d}^{0.6}\exp\left(0.0005\;TOW-1\times 10^{-6}\left(T-10\right)\right)\;+0.8403\;S_{d}^{0.9}\exp(4\times 10^{-5}\;TOW+0.00163\;T)$$
where *r_corr_* is the first-year corrosion rate of metal (μm/y); *T* is the annual average temperature (°C); *P_d_* is the annual average SO_2_ deposition (mg/m^2^d); and *S_d_* is the annual average Cl^−^ deposition (mg/m^2^d).
(10)$$r_{corr}=0.04082\;S_{d}^{0.4411}\exp\left(3.453\;TOW\right)\;[g/m^{2}y]$$
(11)$$r_{corr}=0.3219\;S_{d}^{0.4411}\exp\left(3.453\;TOW\right)\;[\mu m/y]$$

Equation (10) is the formula for the corrosion rate obtained through the calibrated model. However, this equation has units of g/m^2^y; therefore, when this equation is converted to the same units as those in Equation (9), it is equivalent to Equation (11). The noticeable differences observed in the values of each parameter when comparing Equation (9) and the calibrated model (Equation (11)) can be attributed to two reasons. First, Equation (9) takes into account the additional effects of SO_2_ and temperature. Considering these factors, the calibrated model can be improved further. Second, the monitoring data used for calibration reflected the unique climatic characteristics of the Republic of Korea. This point was highlighted by Santana et al. [24] and Corvo et al. [25], who pointed out that ISO-9223 [8] does not accurately account for the climatic peculiarities of specific regions. Consequently, the differences in the parameters between ISO-9223 [8] and the calibrated model used in this study can be attributed to the fact that the calibrated model considered the climatic characteristics specific to the Republic of Korea.

## 4. Conclusions

This study focused on enhancing the atmospheric corrosion model. The improvement process involved developing a physics-based model and calibrating it using monitoring data. First, we assumed the physics-based model to be a nonlinear equation for the TOW and chloride deposition rate. Subsequently, we performed a logarithmic transformation on the nonlinear equation for enhanced computational convenience. The model was then calibrated using the logarithmically transformed linear equation and monitoring data. In this process, a BN was utilized to estimate the parameters based on the dose–response relationship between the physics-based model and the monitoring data. The posterior parameter distributions of the physics-based model differed from the prior distributions due to the varying influence of the corrosion rate on the TOW and the chloride deposition rate.

It is important to acknowledge the limitation of this study, which lies in the assumption of using only two parameters (TOW and chloride deposition rate) for the atmospheric corrosion model. Future research could explore the inclusion of other parameters, such as SO_2_ deposition rate and temperature, to enhance accuracy. The findings of this study provide valuable insights for advancing atmospheric corrosion modeling in various environments.

## Figures and Tables

**Figure 1 materials-16-05326-f001:**
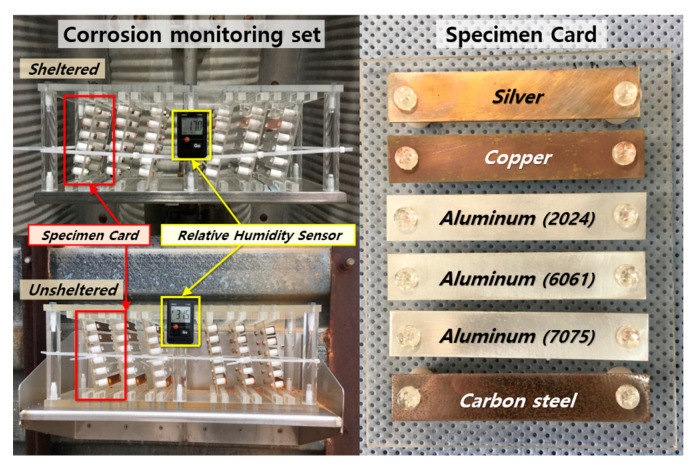
Corrosion-monitoring set and details of a specimen card.

**Figure 2 materials-16-05326-f002:**
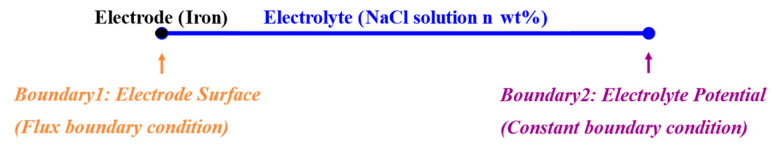
Scheme for physics-based modeling.

**Figure 3 materials-16-05326-f003:**
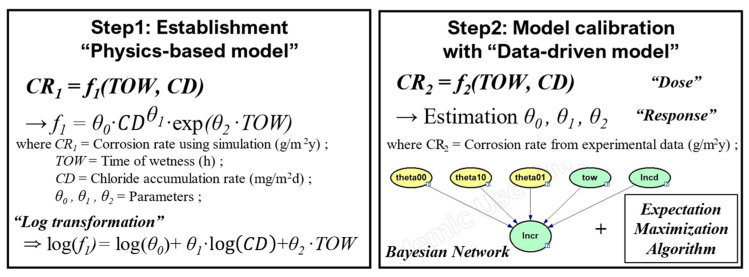
Research methodology.

**Figure 4 materials-16-05326-f004:**
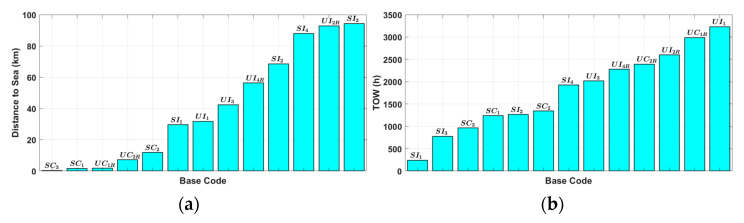
Comparison of atmospheric corrosion monitoring data of the Republic of Korea Air Force showing (**a**) distance to the sea, (**b**) TOW, (**c**) chloride deposition rate, and (**d**) carbon steel corrosion rate.

**Figure 5 materials-16-05326-f005:**
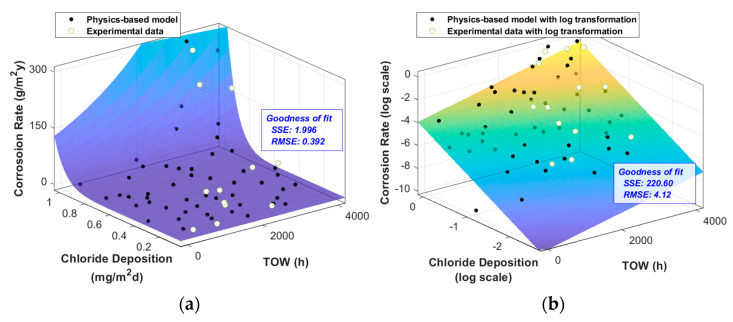
Atmospheric corrosion modeling result: (**a**) raw data, (**b**) with logarithmic transformation.

**Figure 6 materials-16-05326-f006:**
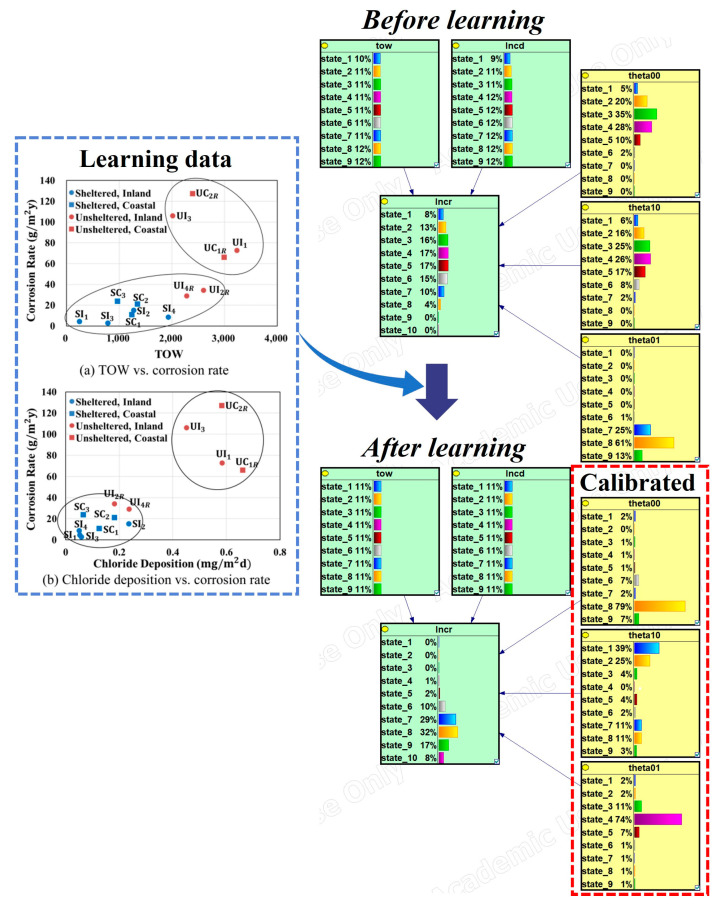
Bayesian network for calibration with the logarithmically transformed form.

**Figure 7 materials-16-05326-f007:**
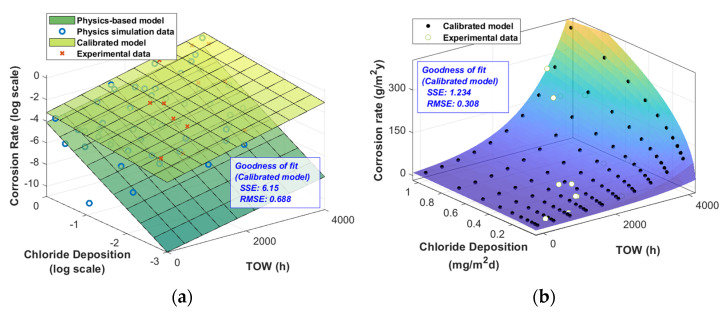
Results of model calibration: (**a**) log scale, (**b**) inverse transformation.

**Table 1 materials-16-05326-t001:** Air bases are categorized by geographical locations and types of aircraft parking areas.

Environment	Coastal	Inland
Unsheltered	${UC}_{1R}$, ${UC}_{2R}$	${UI}_{1}$, ${UI}_{2R}$, ${UI}_{3}$, ${UI}_{4R}$
Sheltered	${SC}_{1}$, ${SC}_{2}$, ${SC}_{3}$	${SI}_{1}$, ${SI}_{2}$, ${SI}_{3}$, ${SI}_{4}$

**Table 2 materials-16-05326-t002:** Parameters for the prior and posterior distributions.

Variable	Distribution	Prior (Mean, SD)	Posterior (Mean, SD)
log(*θ_0_*)	Normal	−4.11, 0.362	−3.1985, 0.6814
*θ_1_*	Normal	2.283, 0.2543	0.4411, 0.4888
*θ_2_*	Normal	3.022, 0.5031	3.4526, 0.5669

**Table 3 materials-16-05326-t003:** Comparison of goodness of fit.

Model	Sum-of-Squares Error	Root-Mean-Squared Error
Physics-based model	1.99587	0.391827
Physics-based model (logarithmic transformation)	220.602	4.11939
Calibrated model (logarithmic transformation)	6.15466	0.688066
Calibrated model	1.23354	0.308038

## Data Availability

Not applicable.

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
