# Peer review of "Physics-Informed, Data-Driven Model for Atmospheric Corrosion of Carbon Steel Using Bayesian Network"

_materials, 2023, doi:10.3390/ma16155326_

Round 1

Reviewer 1 Report

The reviewed paper contains interesting corrosion tests and analyzes using artificial intelligence. Unfortunately, the use of construction materials in the marine environment without protection is not possible due to high corrosion rates. Practically in every case anti-corrosion protection is used in the form of coatings or cathodic protection (structures immersed in water). This information should be on the paper.

Corrosion mechanisms described do not involve corrosion in CO2 which is present in water. It causes a decrease in pH and an increase in corrosion rate.

The failure of a structural element of the helicopter was originally related to damage to the protective organic coating, in addition, this element was made of steel with high mechanical properties, which was relatively sensitive to vibration. This case of damage cannot be related to the performed analyses. Therefore, I propose to delete this part of the work. In general, the research results are of low scientific and practical value.

Author Response

Dear Reviewer,

We appreciate your assistance. According to your comments, the manuscript was revised as a uploaded attachment.

Reviewer 2 Report

This manuscript predicted the corrosion of aircraft structure in atmospheric environment based on physical model and Bayesian network model. The paper was innovative, but lacked some important data. Therefore, it is recommended to be reconsidered after major revisions, here are some of my suggestions:

(1) What is the range of chloride ion concentration in the environmental system and what is the limit concentration?

(2) What is the outdoor environment? This is critical to the aircraft structure corrosion; the manuscript lacks the description of the outdoor environmental conditions.

(3) How do the chloride ions accumulate? Please supplement the experimental data (or other convincing data) related to the process of chloride ion accumulation.

(4) How to verify the accuracy of the models?

Author Response

(The authors gave the same response as above.)

Reviewer 3 Report

It appears that the current calibrated model overestimates the corrosion rate of the material and does not align with the original experimental data. The original data includes manipulated parameters such as coastal, inland, sheltered, unsheltered conditions with and without a roof. The relationship between corrosion rate and time of wetness (TOW) is not solely dependent on sheltered or unsheltered planes, but also on the specific location of the plane. Therefore, a linear model may be suitable for chloride accumulation but not for TOW. The manuscript would benefit from improving the modeling method for TOW.

Reaction 3, the value should be corrected to 4e-.

Line 171, it is not clear whether TOW is related to rainfall. Although the correlation of rainfall with shelters is not shown in Figure 4, it was described in the context.

Line 177 mentions carbon steel.

Line 182 raises the concern that the Latin hypercube technique used to generate samples may not be the most suitable approach in this case, as it assumes equal partitioning of the parameter space. However, this assumption does not hold for the experimental data as the dependencies are not uniform. The studied data includes the influence of various conditions such as coastal, inland, sheltered, unsheltered, with and without a roof.

Figure 6 is missing the label "(b)" for chloride deposition vs. corrosion rate.

Line 213, kindly include the experimental data points in Figure 6 and to show goodness of fit of the model compared to the experimental data.

Author Response

(The authors gave the same response as above.)

Round 2

Reviewer 2 Report

The relevant issues have been resolved. This article can be accepted.